# Diurnal Change of the Photosynthetic Light-Response Curve of Buckbean (*Menyanthes trifoliata*), an Emergent Aquatic Plant

**DOI:** 10.3390/plants11020174

**Published:** 2022-01-10

**Authors:** Azumi Okamoto, Kohei Koyama, Narayan Bhusal

**Affiliations:** 1Department of Agro-Environmental Science, Obihiro University of Agriculture and Veterinary Medicine, Inadacho, Obihiro 080-8555, Japan; 2Department of Agriculture, Forestry and Bioresources, Seoul National University, Seoul 08826, Korea; bhusal.narayan4@gmail.com

**Keywords:** morning reduction, midday depression, midday stomatal closure, aquatic plant, hydrophyte, water plant, emergent plant, *Menyanthes*, wetland, marsh

## Abstract

Understanding plant physiological responses to high temperature is an important concern pertaining to climate change. However, compared with terrestrial plants, information about aquatic plants remains limited. Since the degree of midday depression of photosynthesis under high temperature depends on soil water conditions, it is expected that emergent aquatic plants, for which soil water conditions are always saturated, will show different patterns compared with terrestrial plants. We investigated the diurnal course of the photosynthetic light-response curve and incident light intensity for a freshwater emergent plant, buckbean (*Menyanthes trifoliata* L.; Menyanthaceae) in a cool temperate region. The effect of midday depression was observed only on a very hot day, but not on a moderately hot day, in summer. The diurnal course of photosynthetic light-response curves on this hot day showed that latent morning reduction of photosynthetic capacity started at dawn, preceding the apparent depression around the midday, in agreement with results reported in terrestrial plants. We concluded that (1) midday depression of emergent plants occurs when the stress intensity exceeds the species’ tolerance, and (2) measurements of not only photosynthetic rate under field conditions but also diurnal course of photosynthetic light-response curve are necessary to quantify the effect of midday depression.

## 1. Introduction

Photosynthesis of plants greatly affects global [1,2] and regional [3,4,5,6,7,8,9,10,11,12,13,14,15,16] carbon cycles. Since ecosystem-level photosynthesis is the sum of single-leaf photosynthesis [16,17,18,19,20,21,22], understanding of stress responses of leaves to diurnal and seasonal environmental change is necessary to improve ecosystem carbon cycles modeling [6,8,9,11,12,23,24,25,26,27] and to increase agricultural [22,27,28,29,30,31,32,33,34,35,36,37,38,39,40,41] and forestry [13,42,43] production.

Although photosynthetic rate primarily depends on the incident light intensity [8,10,19,44,45,46], other factors such as air temperature [9], humidity, and vapor pressure deficit also play important roles that impose limitations [10,47,48,49,50,51,52,53]. Midday depression of photosynthesis is a phenomenon where photosynthetic rate decreases during the midday hours when light is not a limiting factor for photosynthesis [6,8,11,21,50,54,55,56,57,58]. The degree of photosynthetic limitation is affected not only by the atmospheric environment surrounding leaves, but also by the soil water content and resultant whole-plant water status [21,31,32,37,38,41,42,43,48,50,54,59,60,61,62,63,64,65]. Thus, it is expected that plants in environments with different water availability will show different degrees of midday depression. Since plants in nature grow in habitats with varying degrees of water availability (i.e., from aquatic environments to deserts), it is important to investigate different plant species from different habitats with varying amounts of water availability.

Emergent plants are a type of aquatic plant where their roots are submerged underwater, and their leaves are above water. This system is unique because the effect of soil water conditions can be excluded from weather conditions; the soils is always water-saturated. However, although much information is available on the midday depression of photosynthesis in rice plants (*Oryza sativa* L.) [11,27,28,66,67], there is limited information on wild aquatic plant species. This situation contrasts with the plentiful information on terrestrial plants in diverse habitats, including both woody species [30,31,37,42,43,54,55,58,61,68,69,70] and herbs [39,56,60,71,72]. Previous studies on midday depression of emergent plants are equivocal. Pearcy et al. [73] reported that a wetland emergent plant species (common reed, *Phragmites australis* (Cav.) Trin. ex Steud. (syn. *Phragmites communis*)) did not show midday depression even on a very hot summer day, when the air temperature exceeded 40 °C. They suggested that transpiration had a cooling effect on the leaves with the availability of abundant water. Similarly, Tsuchiya et al. [74] found no midday depression for the emergent plant, Manchurian wild rice (*Zizania latifolia* (Griseb.) Turcz. ex Stapf.) on a hot summer day. Jones [75] reported that the effect of midday depression in a C_4_ sedge, papyrus (*Cyperus papyrus* L.), was small in a tropical swamp. In contrast to these results, Suyker et al. [4] reported a midday depression of ecosystem carbon uptake on hot days in a boreal fen where some emergent plant species, including buckbean (*Menyanthes trifoliata* L.), dominated. Sale and Orr [76] reported midday depression in an emergent plant, bulrush (*Typha orientalis* Presl.). Additionally, Sale et al. [77] reported reduced photosynthetic efficiency in the afternoon compared with the morning for a floating emergent herb, the common water hyacinth (*Pontederia crassipes* Mart. (syn. *Eichhornia crassipes*)). In support of these leaf-level results, Harenda et al. [8] found that carbon influx on a peatland was affected by photosynthetic limitation caused by a high vapor pressure deficit. Taken together, these results indicate that a midday depression is a common phenomenon under high temperature for both terrestrial and aquatic plants, although further studies are required of emergent aquatic plants.

Recently, Koyama and Takemoto [71] investigated a terrestrial plant species, evening primrose (*Oenothera biennis* L.) by simultaneously measuring the diurnal course of the photosynthetic rate under field light intensity as well as the diurnal course of photosynthetic light-response curve. They found that latent reduction of photosynthetic capacity started at dawn, preceding the apparent midday depression. This indicates that measuring photosynthetic rate under field light intensity alone does not provide sufficient information about midday depression. However, most previous studies of midday depression measured the diurnal course of either the photosynthetic rate realized under actual field light intensity [4,56,59] or the diurnal change of photosynthetic capacity under constant saturating light intensity alone [78,79,80]. Therefore, from the results of these studies it is still unclear to what extent midday depression reduced photosynthetic rate throughout the day, including the morning hours when incident light intensity increased but potential (light-saturated) photosynthetic capacity decreased. Given the fact that water availably differs among terrestrial and aquatic plants, it is necessary to simultaneously investigate the diurnal course of the photosynthetic rate under field conditions and diurnal changes in the photosynthetic light-response curve for aquatic plants. Therefore, the objective of our study was to investigate the diurnal change of the photosynthetic light-response curve of a submerged perennial herb, buckbean (*Menyanthes trifoliata* L.), grew in Hakuryo pond in Obihiro, which is located in a cool temperate region in Hokkaido in Japan.

## 2. Results

### 2.1. Weather Conditions

The measurements were performed twice in summer (July 2021). The first period (18–19 July, hereafter, referred to as “very hot days”) was two successive clear sunny days with very high temperature; the atmospheric temperature (measured at the position of approximate height of the leaves (0.25 m from the ground)) reached 37.9 °C on 19 July on which the first measurement of the diurnal course of photosynthetic light-response curves was performed (Figure 1). The second period (22–23 July, “moderately hot days”) was two successive partially cloudy days with moderate atmospheric temperature; the atmospheric temperature reached only 29.0 °C on 22 July, on which the second measurement of diurnal course of photosynthetic light-response curves was performed (Figure 1). On each of the two-day periods, the diurnal change of photosynthetic light-response curve and atmospheric temperature was measured on one day (19 and 22 July, respectively), and the diurnal change of photosynthetic photon flux density (PPFD) incident on the leaves was measured on the other day (18 and 23 July, respectively). Additionally, the PPFD incident on the horizontal surface above the canopy was measured on all of these days (Figure 1). The data within each period (photosynthetic light-response curve and PPFD incident on the leaves) were combined to estimate diurnal change of net photosynthetic rate on each weather condition (very or moderately hot day). Note that even on hot clear sunny days, the photosynthetic photon flux density (PPFD) incident on the horizontal surface above the *M. trifoliata* stand, measured at a height of 2 m from the ground, declined in the afternoons due to shading by trees at the pond shore. The detailed procedures of the measurements and data analysis are described in the Materials and Methods section.

### 2.2. Photosynthesis

On one of the very hot days (July 19), photosynthetic rate and conductance at high light (PPFD = 1500 μmol m^−2^ s^−1^) decreased from the early morning toward the midday, showed the lowest value at midday when the atmospheric temperature and vapor pressure deficit (VpdL) was highest, and recovered in the evening (Figure 2). Both the light-saturated gross photosynthetic rate (*P*_g_max_) (see Equation (1) in Materials and Methods) and stomatal conductance measured under PPFD = 1500 μmol m^−2^ s^−1^ (*G*_s_PPFD1500_) significantly decreased during midday (10:00–14:00) on the very hot day (generalized linear model, morning vs. midday: *p* < 0.05), but not on the moderately hot day (*P*_g_max_: *p* = 0.734 and *G*_s_PPFD1500_: *p* = 0.324) (Figure 3).

Using data of the diurnal change in PPFD incident on each leaflet (Figure 4, left column) and the observed diurnal change of photosynthetic light-response curves (Figure 2)**,** we estimated the diurnal course of the net photosynthetic rate (*P*_n_) under the actual field light intensity. Midday depression was observed for the estimated actual diurnal courses of net photosynthetic rate (Figure 4, middle column). Additionally, we performed a simulation in which the effect of midday depression was hypothetically excluded. In this simulation, the photosynthetic light-response curve was fixed at that obtained from the earliest measurement of each leaflet within each measurement day. Further, diurnal course of net photosynthetic rate under this hypothetical scenario was then calculated the same way as that of the actual diurnal course of photosynthetic rate as described above, using the same PPFD data for each leaflet. Under this hypothetical situation, no midday depression was simulated (Figure 4, right column). On the moderately hot day (July 22), the leaves showed neither morning reduction of photosynthetic capacity nor midday depression of photosynthetic rate (Figure 3, Figure 5 and Figure 6).

### 2.3. Transpiration

The diurnal courses of the transpiration rate showed quite different patterns from those of photosynthesis or stomatal conductance; transpiration rate did neither show morning reduction nor midday depression even on the hot day (Figure 7 and Figure 8). The diurnal course of the transpiration rate basically followed that of vapor pressure deficit (VPD); transpiration gradually increased from the dawn, reached the highest peak around the midday when the VPD was highest (Figure 2 and Figure 5), and then decreased again in the afternoon.

### 2.4. Daily Carbon Gain

The reduction of daily carbon gain was estimated as the difference between the daily carbon gains under a hypothetical situation (in which the photosynthetic light-response curve was fixed at that obtained on the earliest morning for each leaflet in Figure 2 and Figure 5) and the actual situation (in which the diurnal changes in the photosynthetic light-response curve, as shown in Figure 2 and Figure 5, were taken into consideration), divided by the daily net photosynthesis under the hypothetical situation with no midday depression (Table 1). On a very hot day, daily integrated net photosynthesis of the leaves was reduced by 17.0% (Table 1). No apparent reduction due to midday depression was observed for the moderately hot day (Table 1).

## 3. Discussion

The importance of wetland ecosystems in carbon cycles has long been recognized [4,5,7,8,10,15,81] and wetland ecosystems are highly susceptible to climate change, including temperature increases [82,83]. However, compared with terrestrial plants, quantitative evidence of the effects of high temperature is limited for wild aquatic plants. Especially, information about the midday depression of aquatic plant species is still limited. Therefore, our results provide valuable quantitative empirical evidence. The results of previous studies on emergent plant species are equivocal. Some emergent plant species show midday depression or at least some adverse effect of high temperature or vapor pressure deficit (VPD) on photosynthesis during midday; some examples of such species include rice [27,28,66,67], bulrush [76], water hyacinth [77], and emergent leaves of yellow water lily [46]. However, other species show no visible effect of midday depression (e.g., reed [73], papyrus [75], and Manchurian wild rice [74]). Observations of ecosystem-level carbon flux using the eddy covariance method have shown that wetland ecosystem carbon influx (including paddy rice) is indeed affected by high VPD during midday or in the afternoon [4,8,10,11]. We suggest that the inconsistency among previous studies may result from differences in stress tolerance and stress intensity across different species, climates, and weathers. Our results showed that only on the very hot day did both the photosynthetic capacity and stomatal conductance decrease during midday (Figure 2 and Figure 3) when the VPD was very high (Figure 2). These results indicate that midday depression occurs only when stress intensity exceeds the species’ limits of tolerances. Stress tolerance, especially to high temperature, differs among species. Therefore, if temperature rise due to climate change continues, it is possible that the tolerant species described above (reed, papyrus, Manchurian wild rice, etc.) may show a midday depression in the future.

Our results are consistent with those of Koyama and Takemoto [71] in that information about the actual photosynthetic rate under field conditions as well as the diurnal change in the photosynthetic light-response curve are needed to evaluate the effect of midday depression. On the very hot day, the photosynthetic rate and conductance at high light decreased from the early morning toward midday, showed the lowest value at midday when the atmospheric temperature and VPD were highest, and recovered in the evening (Figure 2). These results are consistent with previous findings for a terrestrial herb, *Oenothera biennis* [71], and some terrestrial woody species [78,79,80] in that latent reduction of the photosynthetic capacity started in the early morning, preceding the apparent midday depression of photosynthesis around midday, which is a phenomenon called the morning reduction of photosynthetic capacity [71]. Investigation of diurnal courses of the photosynthetic rate alone leads to an erroneous conclusion that the depression affects photosynthesis only in the midday hours (Figure 4), though the actual photosynthetic capacity started to decrease at dawn (Figure 2). This suggests that a measurement of the diurnal course of the photosynthetic rate alone does not provide precise information about when reduction starts and to what degree reduction affects daily carbon gain. In most previous studies, midday depression was considered to occur only in the midday hours, ignoring the latent morning reduction of photosynthetic capacity. As a result, the light-saturated photosynthetic rate may have been underestimated in some previous studies if photosynthesis was measured in the late morning on a dry hot day. Furthermore, in most of the previously used methods, it is difficult to quantify the magnitude of reduction of photosynthetic rate. Without information about the diurnal change in the photosynthetic light-response curve, it is difficult to simulate the “hypothetical photosynthetic rate without stress”, as in our study.

The present study has several limitations. First, we did not measure non-stomatal limitation. Midday depression of photosynthesis is caused by both stomatal limitations and non-stomatal limitations such as photoinhibition [56,61,84,85,86,87,88] and reduced Rubisco activation under high temperature [89,90]. Our findings do not preclude the possibility of non-stomatal limitation influencing photosynthesis. Second, we ignored photosynthetic induction time. During the measurements of photosynthetic light-response curves, on each occasion of changing light intensity, we waited until equilibrium and measured photosynthetic rate. In reality, leaves experience rapidly fluctuating light intensity [19,29,91,92,93,94]; therefore, ignoring mesophyll and biochemical limitations under fluctuating light [95,96,97,98,99,100] could result in overestimation of the photosynthetic rate. Third, we used a red/blue light-emitting diode (LED) light source to measure the photosynthetic light-response curves. Thus, the magnitude of heat load [101,102,103] may be different from that caused by natural sunlight. Fourth, although the leaves of this emergent plant species exist in the air, the water temperature surrounding the roots may also affect whole-plant physiology [104]; however, the effect of water environment was not investigated in the present study. Finally, we only investigated one species from one climate; however, the stress responses and degree of photosynthetic limitation vary among different species [32,37,105,106], cultivars [30,31,35,64,107], and plants in different growth conditions [42,66]. Furthermore, even within a single site, causes of photosynthetic limitation change seasonally [9,13,42] and annually [70]. Therefore, further studies that consider these factors and include various species and environments are needed before generalization of the present results.

## 4. Materials and Methods

### 4.1. Study Site

We conducted this study in Hakuryo pond located on the campus of Obihiro University of Agriculture and Veterinary Medicine (45°52′ N 143°10′ E, altitude: 79 m a.s.l.) in Hokkaido, Japan. The mean annual temperature and precipitation at the Obihiro Weather Station, which is within 10 km from the site, during 1998–2017 were 7.2 °C and 937 mm, respectively [108].

### 4.2. Plant Materials

Buckbean, or bogbean (*Menyanthes trifoliata* L.; Menyanthaceae), is a freshwater emergent aquatic perennial herb with trifoliate leaves. It is distributed throughout the northern hemisphere [109], including East Asia [110,111,112,113,114,115], Europe [116,117], and North America [3,4,5]. Although the origin of *M. trifoliata* at the study site was unknown (i.e., whether it was artificially introduced or naturally dispersed), according to a photographic record (Sato M, unpublished pictures), the plants have been growing at the study site under natural conditions for at least 15 years. Three healthy, undamaged leaflets from three leaves (labeled Leaflets #5, #8, and #9, respectively) were selected from three different individual ramets. Prior to the measurements, we constructed a small temporary wooden pier to access these leaves. The pier was carefully constructed so as not to damage the plant materials. Before measuring, we marked part of lamina of each sample leaflet with a red pen, so as to repeatedly measure the same position on the same leaflet throughout the measurement period.

### 4.3. Field Measurements

Measurements were performed twice in July 2021, following the procedure described in [71] with some modifications. The first period (18–19 July, “very hot days”) was two successive clear sunny days with very high temperature; the atmospheric temperature at 0.25 m from the ground reached 37.9 °C on 19 July (Figure 1). The second period (22–23 July, “moderately hot days”) was two successive partially cloudy days; the atmospheric temperature reached only 29.0 °C on 22 July (Figure 1). In each two-day period, the diurnal change of photosynthetic light-response curve was measured on one day (19 and 22 July), and the diurnal change of photosynthetic photon flux density (PPFD) incident on the leaves was measured on the other day (18 and 23 July) as described below.

### 4.4. Photosynthetic Measurements

Photosynthetic light-response curves were measured with a portable photosynthesis system (LI-6400; LI-COR, Lincoln, NE, USA) equipped with an LI-6400-02B red/blue LED light source (the peak wavelengths: 665 nm (red) and 470 nm (blue)). On each measurement day (19 or 22 July), photosynthetic light-response curve was repeatedly measured for the three leaves during 6:00–19:00. Since we used only one LI-6400 portable photosynthetic system, we measured the three leaflets in turn (e.g., leaflets #5-#6-#8-#5-#6-#8-…, etc.). Leaflet #8 at around 9 am on 19 July was inadvertently not measured due to an operational mistake. At each curve measurement, we first induced the leaflet by keeping incident PPFD on the leaflet at 1000–1500 μmol m^−2^ s^−1^ until equilibration. The induction was omitted if the incident natural sunlight level was high at that moment. After that, we progressively lowered the incident PPFD on the leaflet surface ((2000), 1500, 1000, 750, 500, 250, 125, 63, 32, and 0 μmol m^−2^ s^−1^). Measurement under PPFD = 2000 μmol m^−2^ s^−1^ was not performed on the first measurement (19 July) because the LED light source we used on the day was unable to supply PPFD > 1800 μmol m^−2^ s^−1^, due to aging of the LED. On the second measurement day (22 July), photosynthetic rate under PPFD = 2000 μmol m^−2^ s^−1^ was successfully measured using another LI-6400-02B light source. We observed that the leaves were nearly light-saturated around PPFD = 1500 μmol m^−2^ s^−1^ (Figure 2), indicating that photosynthetic light-response curve parameters can be estimated without the data of PPFD = 2000 at least for this species. On each occasion of changing light intensity, we kept the PPFD constant until the equilibration of the leaves. The CO_2_ concentration of the air entering the LI-6400 chamber was controlled at 400 ppm. Diurnal change of temperature and humidity of the air surrounding the leaves was measured with a thermo-hygrometer (TT-492, Tanita, Tokyo, Japan), which was hanged below the seat of small plastic chair put on the shore near the plants, and was set at the position of the approximate height of the leaves (0.25 m from the ground). During the daytime hours, the thermo-hygrometer was shaded by a parasol or a picnic sheet so as not to receive direct sunlight while allowing the natural ventilation. The temperature of the air flow into the LI-6400 chamber was controlled automatically using the air conditioner of LI-6400 so as to trace the diurnal course of atmospheric temperature measured at the external thermo-hygrometer. The humidity of the air flow into the LI-6400 chamber was controlled manually by adjusting the desiccant valve on the LI-6400 so as to trace the diurnal course of atmospheric humidity measured at the thermo-hygrometer. Prior to each measurement of light-response curve, the external air temperature and humidity were recorded using the thermo-hygrometer described above, and for simplification, during single measurement of light-response curve, the condition of the air flow into the chamber was fixed at these values. Additionally, during daytime, the chamber and console units of LI-6400 were occasionally shaded by a small umbrella to avoid excessive heating by sunlight.

### 4.5. PPFD Measurements

On each measurement day (18 or 23 July), incident PPFD was measured with quantum sensors (MIJ-14PAR) fixed on the top of poles. Each sensor was set at the height of each target leaflet lamina, and was inclined to measure the incident PPFD on the inclined surface of the lamina [19,71,92]. Each sensor was connected to a voltage logger (LR5041; HIOKI, Ueda, Japan) and the voltages were recorded every 10 min on each measurement day. These values were transformed into PPFD using sensor-specific coefficients.

### 4.6. Data Analysis

Statistical analyses were performed using R ver. 4.12 [118] software with the packages “ggbeeswarm” [119], “ggplot2” [120], “ggpubr” [121], and “minpack.lm” [122]. The net photosynthetic rate per unit area of leaf (*P*_n_ μmol m^−2^ s^−1^) was assumed to be expressed by a nonrectangular hyperbola [19,123,124]:(1)$$P_{n}=\frac{\phi I+P_{g\_\max}-\sqrt{{\left(\phi I+P_{g\_\max}\right)}^{2}-4\theta \phi IP_{g\_\max}}}{2\theta }-R,$$
where *I* (μmol quanta m^−2^ s^−1^) indicates incident PPFD on each leaflet at each moment, and *P*_g_max_ (μmol m^−2^ s^−1^) indicates maximum gross photosynthetic rate when *I* approaches infinity. *ϕ* (mol CO_2_ mol^−1^ quanta) and *θ* (dimensionless) indicate the initial slope and the convexity, respectively. *R* (μmol m^−2^ s^−1^) indicates dark respiration rate. These parameters were fitted with the Levenberg-Marquardt algorithm using the function *nls.lm* [122]. To test the difference in the light-saturated gross photosynthetic rate (*P*_g_max_) and stomatal conductance between early morning and during daytime hours on each day, a generalized linear model was constructed using the R function *glm* (family = Gamma (link = “log”)); the values obtained in the earliest morning were compared with the lowest value observed during the midday hours (10:00–14:00) on each day. We used a Gamma error distribution because it is used to describe continuous and positive variables [125,126].

Diurnal changes in the curve parameters between two successive measurements within each day were estimated by interpolating these parameters every 10 min. The parameters before the first measurements and after the last measurements within each day were assumed to be constants, fixed at the values of the first- and the last measurements on each day, respectively. In each two-day period, diurnal change of photosynthetic light-response curve for every 10 min was estimated for one day as described above, and diurnal change of incident light PPFD for every 10 min was measured on the other day as described in the previous subsection. Since the weather condition of the two days within each period was similar (Figure 1), we assumed that incident light intensity on study leaves were similar within each period. Then, we combined the data within each period to estimate diurnal change of photosynthetic rates for each one day from each of the weather conditions (i.e., the very hot clear sunny day (19 July) or the moderately hot day (22 July)). Daily integrated PPFD incident on the leaves and net photosynthesis for each weather condition were calculated by adding these instantaneous values for 24 h (i.e., including nighttime respiration), based on the assumption that these instantaneous rates were constant within each of the 10-min intervals.

The transpiration rate per unit area of leaf (*T*_r_ mmol H_2_O m^−2^ s^−1^) was calculated using the same procedure as that for the net photosynthetic rate with the modification that, instead of the nonrectangular hyperbola (Equation (1)), the empirical rectangular hyperbola [47,71] was fitted to the observed PPFD-*T*_r_ relation as follows:(2)$$T_{r}=\frac{b_{1}b_{2}\left(I-q\right)}{b_{1}+b_{2}\left(I-q\right)},$$
where *q*, *b*_1_ and *b*_2_ are empirical parameters fitted with the function *nls.lm* [122].

### 4.7. Simulation

To evaluate the effect midday depression on daily carbon gain, we estimated photosynthetic rate under a hypothetical situation in which the effect of midday depression was excluded. Under this simulation, we fixed the photosynthetic light-response curve parameters at the values that were obtained from the earliest measurements in the morning [68,71]. Then, daily photosynthetic rate was calculated in the same way as described in the previous subsection.

## Figures and Tables

**Figure 1 plants-11-00174-f001:**
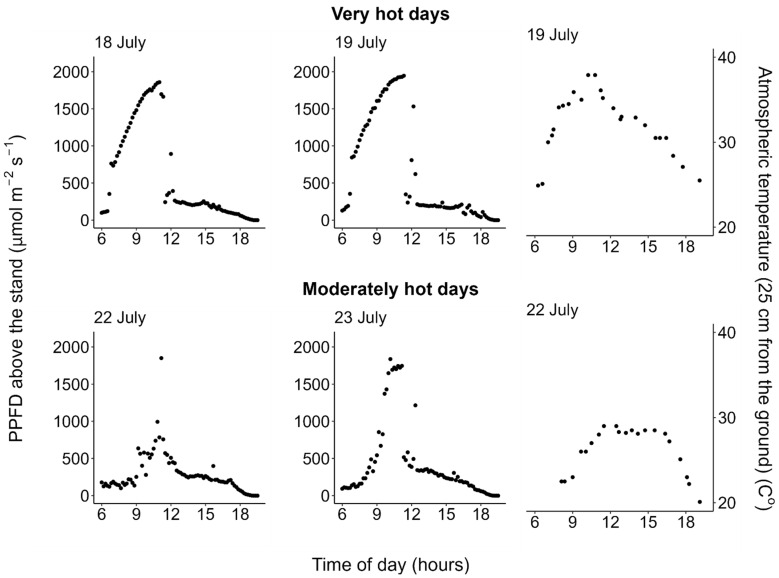
Diurnal courses of photosynthetic photon flux density (PPFD) incident on the horizontal surface above the *Menyanthes trifoliata* stand, measured at the height of 2 m from the ground, and of atmospheric temperature measured at the position of the approximate height of the leaves (0.25 m from the ground). Note that even on the hot clear sunny days, incident PPFD above the stand declined in the afternoons because of shading by the trees at the pond shore. All data are available in the Appendix A.

**Figure 2 plants-11-00174-f002:**
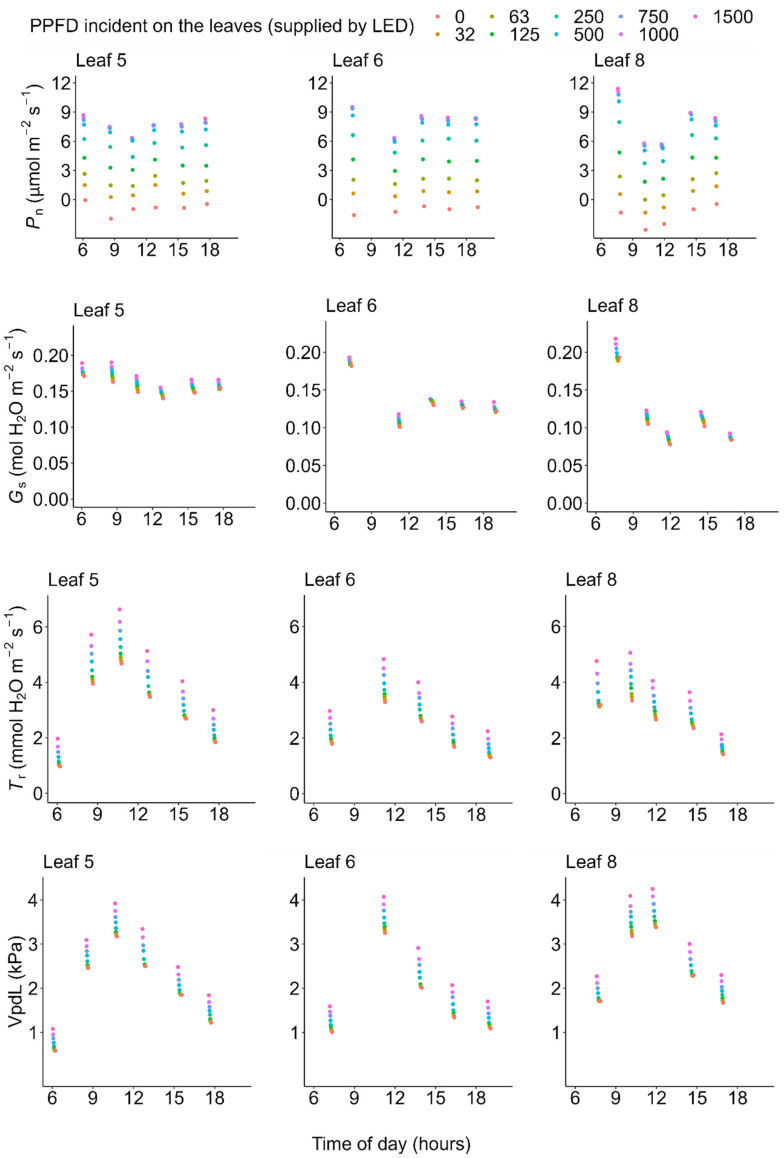
Diurnal courses of the light-response curves on a very hot day (19 July 2021). PPFD: photosynthetic photon flux density supplied with a light-emitting diode (LED) light source (LI-6400-02B), *P*_n_: net photosynthetic rate, *G*_s_: stomatal conductance, *T*_r_: transpiration rate, and VpdL: vapor pressure deficit based on leaflet temperature. These values were measured with an LI-6400 system. All of the data are available in the Appendix A.

**Figure 3 plants-11-00174-f003:**
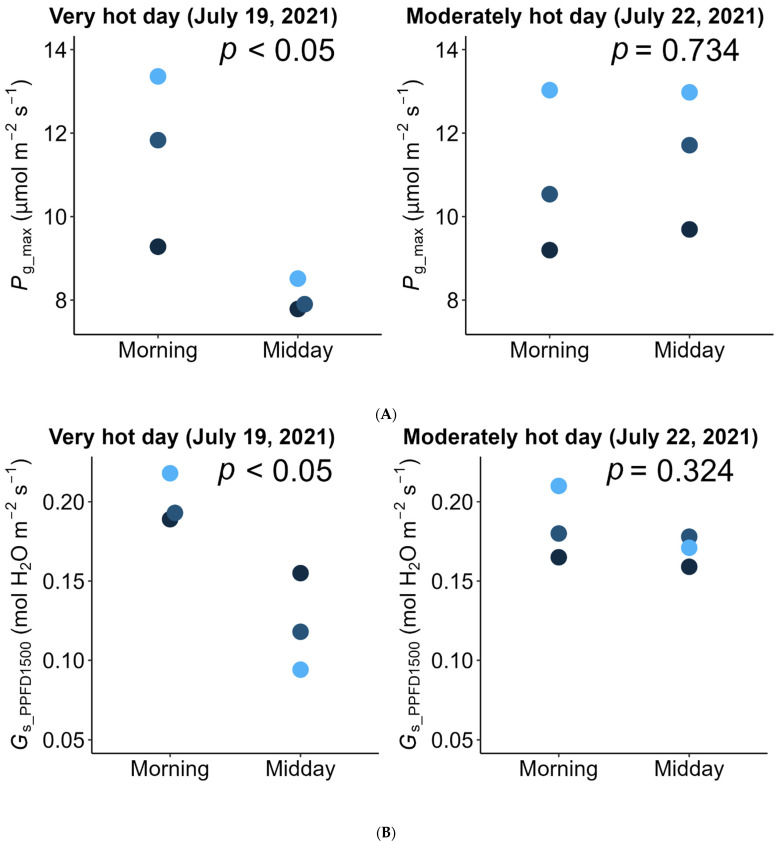
(**A**) Light-saturated gross photosynthetic rate (*P*_g_max_) (see non-rectangular hyperbola (Equation (1)) in the Materials and Methods) and (**B**) stomatal conductance measured under PPFD = 1500 μmol m^−2^ s^−1^ (*G*_s_PPFD1500_) observed in the morning (i.e., the earliest measurement on each day) vs. midday (i.e., the lowest values during 10:00–14:00). The results for the same leaflet measured at different times or days are shown in the same color. The *p*-values shown on the panels are the results of the generalized linear models to determine whether the differences (morning vs. midday) were significant.

**Figure 4 plants-11-00174-f004:**
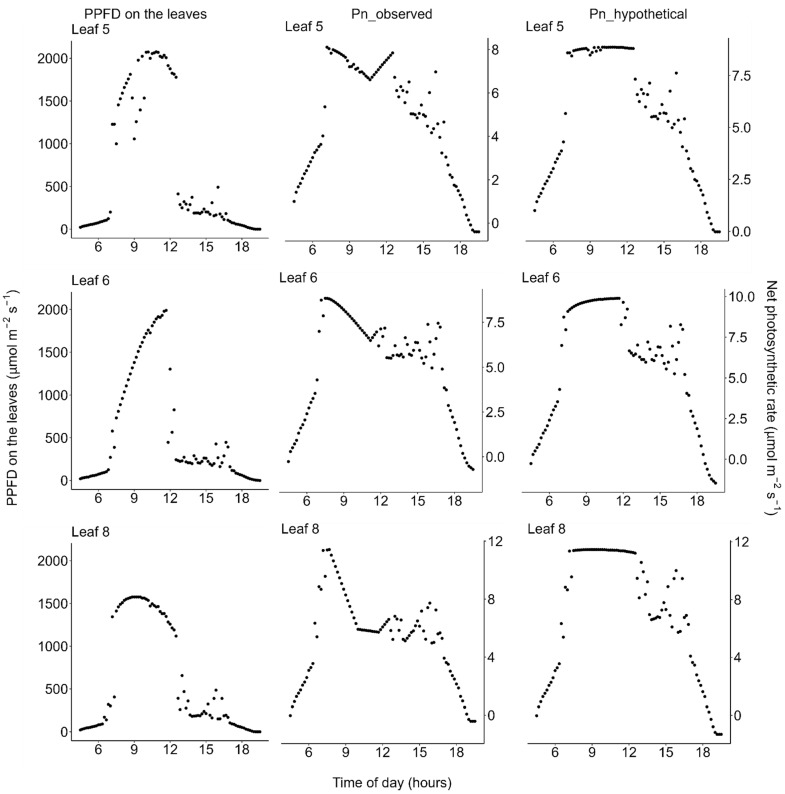
Diurnal courses on the very hot days (18 and 19 July 2021). (**Left**) Photosynthetic photon flux density (PPFD) incident on each leaflet of *Menyanthes trifoliata*, (**middle**) *P*_n_observed_: actual net photosynthetic rate calculated based on both the observed diurnal change of the photosynthetic light-response curve and the diurnal change of PPFD, and (**right**) *P*_n_hypothetical_: simulated hypothetical net photosynthetic rate calculated by holding the photosynthetic light-response curve constant throughout the day, fixed at the curve observed in the early morning for each leaflet. Note that: (1) PPFD was measured on 18 July and the PPFD data were used to estimate photosynthesis on 19 July, the day on which the light-response curves shown in Figure 2 were measured; and (2) in the afternoon, the plants were shaded by trees on the pond shore (see Figure 1 for detailed weather conditions). All data are available in the Appendix A.

**Figure 5 plants-11-00174-f005:**
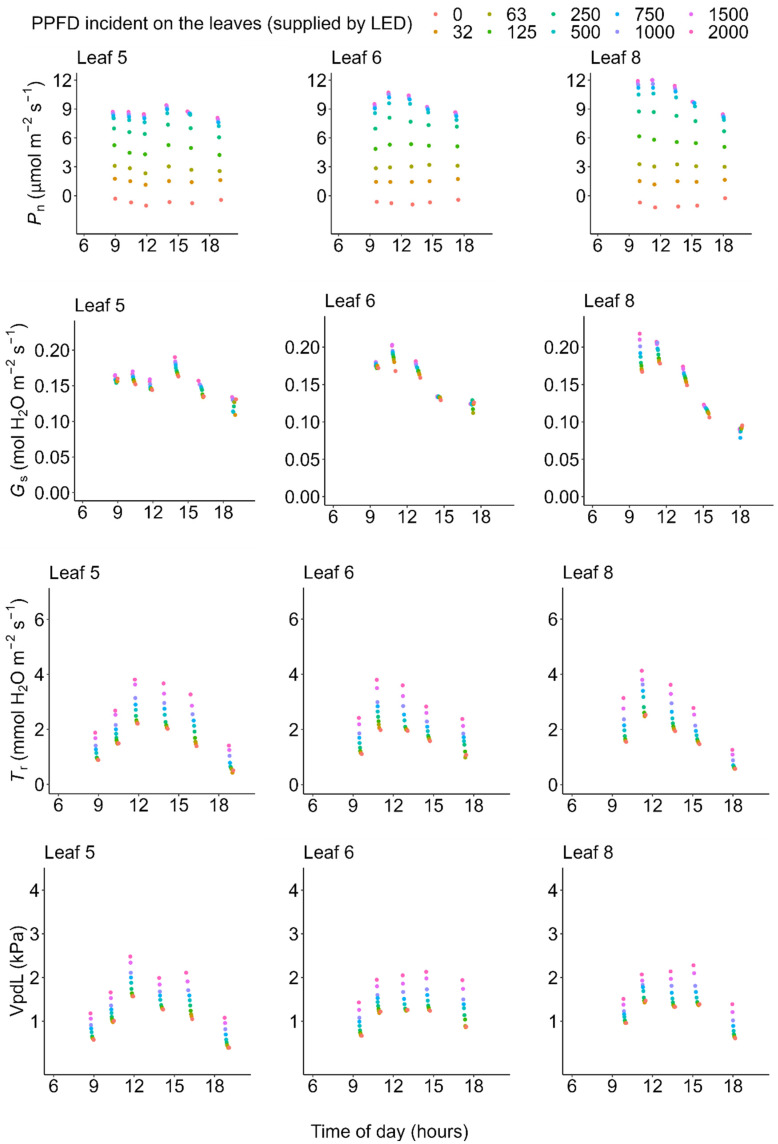
Diurnal courses of light-response curves on a moderately hot day (22 July 2021). PPFD: photosynthetic photon flux density supplied with an LED light source (LI-6400-02B), *P*_n_: net photosynthetic rate, *G*_s_: stomatal conductance, *T*_r_: transpiration rate, and VpdL: vapor pressure deficit based on leaflet temperature. These values were measured with an LI-6400 system. All data are available in the Appendix A.

**Figure 6 plants-11-00174-f006:**
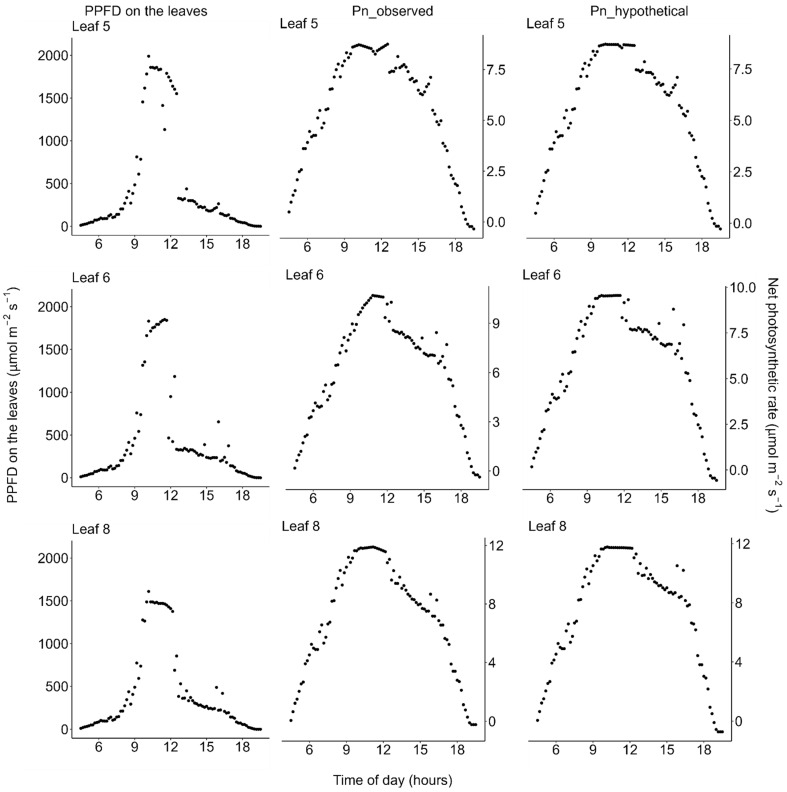
Diurnal courses of photosynthesis on the moderately hot days (22 and 23 July 2021). (**Left**) Photosynthetic photon flux density (PPFD) incident on each leaflet of *Menyanthes trifoliata*, (**middle**) *P*_n_observed_: actual net photosynthetic rate calculated based on both the observed diurnal change of photosynthetic light-response curve (Figure 5) and the diurnal change of PPFD, and (**right**) *P*_n_hypothetical_: simulated hypothetical net photosynthetic rate calculated by holding the photosynthetic light-response curve constant throughout the day, fixed at the curve observed in the early morning for each leaflet (Figure 5). Note that: (1) PPFD was measured on 23 July and the PPFD data were used to estimate photosynthesis on 22 July, the day on which the light-response curves shown in Figure 5 were measured, and (2) in the afternoon, the plants were shaded by trees on the pond shore (see Figure 1 for detailed weather conditions). All data are available in the Appendix A.

**Figure 7 plants-11-00174-f007:**
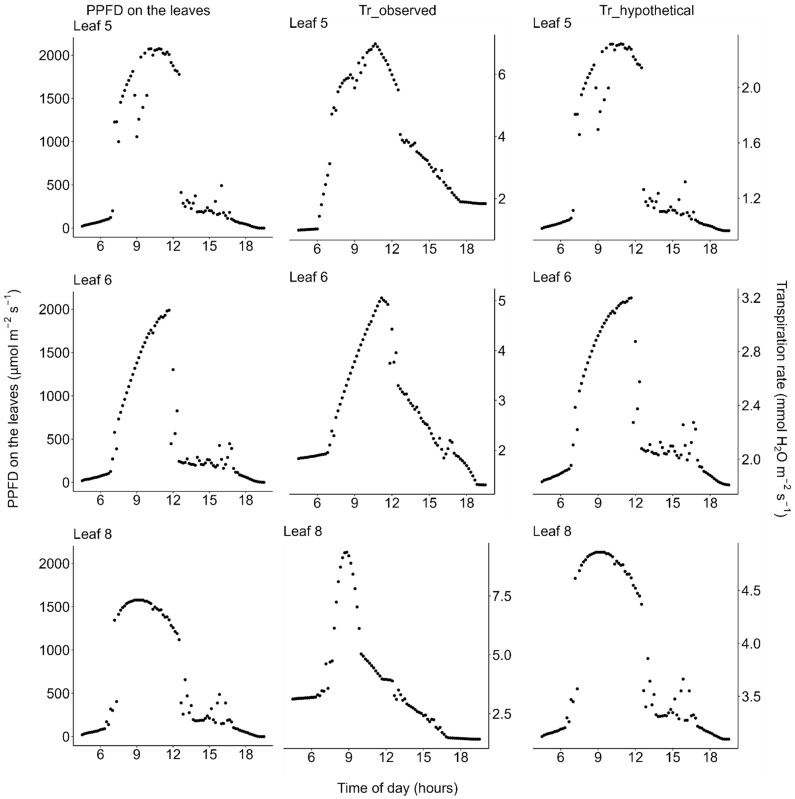
Diurnal courses of transpiration on the very hot days (18 and 19 July 2021). (**Left**) Photosynthetic photon flux density (PPFD) incident on each leaflet of *Menyanthes trifoliata*, (**middle**) *T*_r_observed_: actual transpiration rate per unit area of leaf calculated based on both the observed diurnal change of PPFD-transpiration response curve (Figure 2) and the diurnal change of PPFD, and (**right**) *T*_r_hypothetical_: simulated transpiration rate calculated by holding the PPFD-transpiration response curve constant throughout the day, fixed at the curve observed in the early morning for each leaflet (Figure 2). Note that: (1) PPFD was measured on 18 July and the PPFD data were used to estimate transpiration on 19 July, the day on which the light-response curves shown in Figure 2 were measured, and (2) in the afternoon, the plants were shaded by trees on the pond shore (see Figure 1 for detailed weather conditions). All data are available in the Appendix A.

**Figure 8 plants-11-00174-f008:**
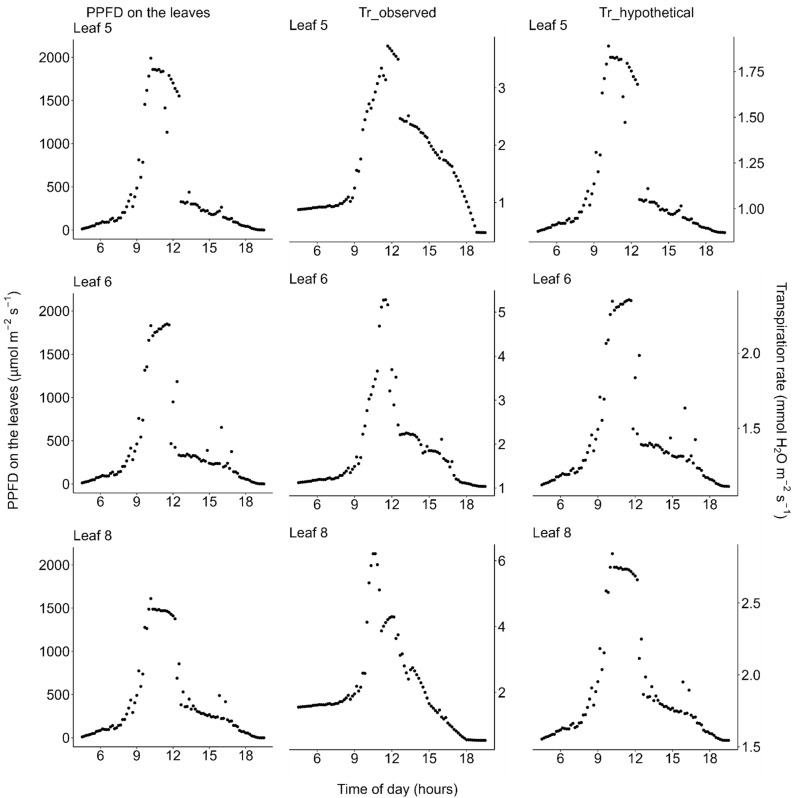
Diurnal courses of transpiration on the moderately hot days (22 and 23 July 2021). (**Left**) Photosynthetic photon flux density (PPFD) incident on each leaflet of *Menyanthes trifoliata*, (**middle**) *T*_r_observed_: actual transpiration rate per unit area of leaf calculated based on both the observed diurnal change of the PPFD-transpiration response curve (Figure 5) and the diurnal change of PPFD, and (**right**) *T*_r_hypothetical_: simulated transpiration rate calculated by holding the PPFD-transpiration response curve constant throughout the day, fixed at the curve observed in the early morning for each leaflet (Figure 5). Note that: (1) PPFD was measured on 23 July and the PPFD data were used to estimate transpiration on 22 July, the day on which light-response curves shown in Figure 5 were measured; and (2) in the afternoon, the plants were shaded by trees on the pond shore (see Figure 1 for detailed weather conditions). All data are available in the Appendix A.

**Table 1 plants-11-00174-t001:** Daily integrated photosynthesis.

	Unit	Median Value (Range)	
		Very Hot Day (19 July)	Moderately Hot Day (22 July)
Daily light integral incident on the inclined leaflet surfaces	mol photon m^−2^ leaf day^−1^	34.7 (31.1–39.2)	24.8 (23.9–26.2)
Actual daily integrated net photosynthesis (A)	mol CO_2_ m^−2^ leaf day^−1^	0.255 (0.245–0.263)	0.332 (0.295–0.375)
Simulated daily integrated net photosynthesis (with no midday depression) (B)		0.309 (0.276–0.345)	0.329 (0.299–0.383)
Reduction ^1^: (B − A)/B (%)	%	17.0% (11.2–23.6%)	−1.0% (−6.5–2.1%)

^1^ The reduction was calculated for each leaflet, and the mean value was shown in the table. This mean value is not exactly equal to that calculated as the difference of the mean values of A and B divided by the mean value of B.

## Data Availability

All data presented in this article, including the LI-6400 data (e.g., photosynthesis, stomatal conductance, transpiration, and temperature inside the chamber) and the data obtained with the light sensors, are available in the Appendix A.

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
