# Peer review of "Diurnal Change of the Photosynthetic Light-Response Curve of Buckbean (Menyanthes trifoliata), an Emergent Aquatic Plant"

_plants, 2022, doi:10.3390/plants11020174_

Round 1

Reviewer 1 Report

The manuscript by Okamoto et al. is devoted to an important problem: the relationship between midday depression of photosynthesis and the environmental conditions, particularly, the atmospheric temperature. This work contains very interesting results, but I have some comments to the manuscript:

  1. P. 1, line 28: not “plans”, but “plants”?
  2. P. 2, line 56: not “transpration”, but “transpiration”?
  3. P. 4, lines 118-121: “These results were consistent with the previous reports that…”. More appropriate for the "Discussion" section.
  4. Supplementary: It is better to explain the column names in the supplementary file. This will make it easier to use the document without referring to the article often.
  5. P. 9, line 182; p. 10, line 192: not “rat”, but “rate”?
  6. Discussion: The Discussion section should be expanded by analyzing the results of the work in more detail. In particular, a discussion of the results on transpiration rate and stomatal conductance will make this section more interesting and complete.
  7. P. 11, lines 208-209: “The present results of the aquatic plant M. trifoliata are generally in agreement with the results of a terrestrial herb, Oenothera biennis [63].”: Please add very brief information about the results of work [63] to make this section comfortable to read (without referring to the Introduction section or the original work).
  8. P. 11, lines 202-208: From my point of view, diurnal courses of photosynthetic rate and diurnal change of photosynthetic light-response curve are consistent with each other, and the diurnal courses of photosynthetic rate show the start of the latent morning reduction of photosynthetic capacity before the midday depression. I think that this approach allows one to more accurately assess the degree of reduction of photosynthetic capacity, but does not reveal any new processes that could not be seen by analyzing only the rate of photosynthesis under filed conditions during the day.
  9. P. 13, lines 329-336: Please give a brief description of this procedure to make this section reader-friendly.

Author Response

Please find the Author's Reply to the Review Report in the PDF file.

Reviewer 2 Report

This study describes the difference of the photosynthetic rate of an emergent plant species on a hot vs less hot day. The authors find that midday depression is only observed under the hot day condition. Similar studies in the literature are few, therefore this study has merit and I find that the results are appropriate for the journal “Plants”.

A few sections could be improved and this includes the figure presentation which is extensive and hard to read/understand because single measurements are displayed instead of average values. As such no statistics and significance is included, which should be required for a scientific publication.

In addition, the discussion is very short. Perhaps comment and discuss more on the following points:

  • No midday depression, but diurnal profile, perhaps discuss the difference of depression and maximum of photosynthesis
  • Include references and discussion of references from the Introduction e.g. about the other water loving plants (rice and wild rice)
  • How important is the aquatic environment with regards to midday depression or maximum?

Finally, it bothers me that there is no single light response curve (in the classical sense) displayed in the manuscript even though the wording is included in the title. Perhaps, change the title to “Diurnal change of photosynthetic rate in the buckbean....”

Figure 1:

In order to better visualize how PPFD and temperature correlated:

Combine the plots for PPFD and temperature for 19 July and possibly also 18 July

Combine the plots for PPFD and temperature for 22 July and possibly also 23 July

The last part of the figure legend (lines 111-113) should be moved (or added) to the Results

Figure 2:

Calculate the average value and variation for the leaves instead of displaying all the individual values.

The midday depression is most apparent for high PPFD values, why is the low PPFD measurements also included? Is that necessary?

Also consider to connect the dots, or add an arrow, so it is evident that the Pn value is going down.

It looks like the measurements were not taken at exactly the same time for each leaf, when calculating the average, calculate the average time point as well.

Why is time point 9 missing for leaf 6?

Consider changing the time scale (x axis) to diurnal scale, where time 0 is lights on (sunrise)

Figures 3, 5, 6 and 7:

Since the pattern for each leaf is similar, display the average value instead of single values per leaf

Figure 4:

Similar comments as for Figure 2

Other minor comments:

Line 30

“Understanding”

48

“the soil is always”

68 and 77

Delete “realized”

141

“course”

147

“On the very hot day”

148

Describe how the percentages were calculated, add reference to the Methods and display the values in table format

207

photosynthetic capacity under what?

211

Replace “price” with “valuable”

261

What color light was the LED? Add nm description

Author Response

(The authors gave the same response as above.)

Reviewer 3 Report

The authors submitted a manuscript entitle “Diurnal change of photosynthetic light-response curve of the buckbean (Menyanthes trifoliata), an emergent aquatic plant” to Plants. They collected photosynthesis, transpiration and daily carbon gain indicators from field grow buckbean plants in very hot day and moderately hot day.

I don't think this is a well-designed and well-prepared study .

Buckbean is an aquatic plant; I think they should provide the water temperature.  

The authors selected three healthy, undamaged leaflets from three leaves from three different individual ramets; I think they should select more samples.

They used very hot day and moderately hot day, these words were too vague.

The authors should perform statistical methods to support their results.

Round 2

Reviewer 1 Report

Dear Authors,

Thank you for your answers. I recommend the article for publication.

Author Response

Author's Reply to the Review Report (Reviewer 1, Round 2)

We appreciate your time and careful review of our manuscript. Thank you very much.

Kohei Koyama